# Human Monkeypox Experience in a Tertiary Level Hospital in Milan, Italy, between May and October 2022: Epidemiological Features and Clinical Characteristics

**DOI:** 10.3390/v15030667

**Published:** 2023-03-02

**Authors:** Caterina Candela, Angelo Roberto Raccagni, Elena Bruzzesi, Costanza Bertoni, Alberto Rizzo, Gloria Gagliardi, Diana Canetti, Nicola Gianotti, Davide Mileto, Maria Rita Gismondo, Antonella Castagna, Silvia Nozza

**Affiliations:** 1Infectious and Tropical Diseases Department, Vita-Salute San Raffaele University, 20127 Milan, Italy; 2Laboratory of Clinical Microbiology, Virology and Bioemergencies, Sacco Hospital, 20900 Milan, Italy; 3Infectious Diseases Unit, IRCCS San Raffaele Scientific Institute, 20097 Milan, Italy

**Keywords:** *Monkeypox virus*, mpox infection, men who have sex with men, sexually transmitted infections, global outbreak

## Abstract

Background: *Monkeypox virus* (mpxv) started to spread to Europe and North America at the beginning of the current outbreak in May 2022, and the World Health Organization (WHO) declared Human Monkeypox (mpox) as a public health emergency of international concern (PHEIC) in July 2022. The aim of this observational analysis is to describe demographical data, symptoms presentation and clinical course till outcome of individuals diagnosed with mpox, between May and October 2022, at our open-access Sexual Health Clinic in IRCCS San Raffaele Hospital in Milan, Italy. Methods: Among people who accessed our Sexual Health Clinic, we considered, as suspected diagnosis of mpox, individuals with consistent symptoms and epidemiological criteria. Following the physical examination, oropharyngeal, anal, genital and cutaneous swabs, plus plasma, urine and seminal fluid were collected as biological materials to detect mpxv DNA. We also performed a screening for sexually transmitted infections (STIs). Results: Overall, 140 individuals with mpox were included in this study. Median age was 37 (interquartile, IQR 33, 43) years old. Males were 137 (98%) and men who have sex with men (MSM) were 134 (96%). As risk factors, we detected travels abroad in 35 (25%) individuals and close contact with mpox cases in 49 (35%). There were 66 (47%) people living with HIV (PLWH). Most frequent symptoms were fever (59%), lymphadenopathy (57%), cutaneous (77%), genital (42%), anal (34%) and oral (26%) lesions, proctitis (39%), sore throat (22%) and generalized rash (5%). At mpox diagnosis, we also observed *N. gonorrhoeae* in 18 (13%) cases, syphilis in 14 (10%) and *C. trachomatis* in 12 (9%). Two (1%) people received a concomitant diagnosis of HIV infection. We attended to 21 (15%) complications, with nine (6%) cases of hospitalization including six (IQR 3,7) median hospital days. Forty-five (32%) patients were treated with non-steroidal anti-inflammatory drugs (NSAIDs), 37 (26%) with antibiotics and eight (6%) with antiviral drugs. Conclusions: Similarly to other international cohorts, sexual transmission was most frequently present, and concomitant STIs were common. Symptoms were heterogenous, self-resolving and responsive to therapy. Hospitalization was necessary in few patients. There is uncertainty about the future development of mpox and further studies (e.g., potential disease reservoirs, other possible means of transmission, predictors of severe disease) are still needed.

## 1. Introduction

In May 2022, the outbreak of human monkeypox (mpox) started to spread to Europe and North America and in July 2022 the World Health Organization (WHO) declared it as a public health emergency of international concern (PHEIC) [1,2,3,4]. Today, more than 85,900 confirmed total cases of monkeypox virus (mpxv) infections have been counted, with 97 deaths, across 110 countries [5]. Specifically, the 10 most affected countries are: United States of America, Brazil, Spain, France, Colombia, United Kingdom, Mexico, Peru, Germany and Canada. Together, these countries account for 85.4% of the total cases reported. The global trend of new cases increased by 69.8% in week 6–12 February (n = 270 cases) compared to week 30 January–5 February (n = 159 cases). The majority of cases in the past 4 weeks were notified from the Region of the Americas and Africa [5].

Mpxv is an enveloped double-stranded DNA virus that belongs to the Orthopoxvirus genus, Poxviridae family [6]. There are two distinct strains of mpxv: clade one (I) has been responsible for disease in the Congo Basin and Central Africa, while clade two (II) has been isolated in West Africa [7]. Precisely, clade II consists of two subclades, IIa and IIb, is less virulent, and lacks several genes present in strain I. Global notifications suggest that the current mpox epidemic seems to have originated from the West Africa clade [8].

Various animal species have been identified as susceptible to mpxv, such as squirrels, rodents and monkeys [8], but the natural history of mpox remains uncertain and further studies are needed to identify the exact reservoir(s) and the way mpxv circulation is maintained in nature [9,10].

Initially, mpxv was identified as a cause of disease in humans in the 1970s in the Democratic Republic of the Congo [11,12] and subsequently other cases occurred in Central and West Africa. The first outbreak of mpox in the Western-hemisphere took place in the United States in 2003 [13], followed by sporadic cases in nonendemic countries, mostly related to travel [14,15,16,17]. The first case of mpox in this current global outbreak was recognized in the United Kingdom in mid-May 2022 and, since then, other cases have been reported worldwide [1,2,3,4]. On the contrary, all of these were not associated with recent travel to endemic areas or close contact with a person known to have mpox, providing evidence of community spread [9,10].

Close contact with infectious material from cutaneous and/or mucosal lesions and bodily fluids is considered the key pathway for human-to-human transmission. Sexual activity has been well established as the main risk factor. Many early cases of mpox involved patients who had joined in the International Pride Parade on the Island of Gran Canarias, Spain, linked to transmission chains in several European countries [1,2,3,18,19,20,21,22]. Some patients also reported having multiple sexual partners in the previous weeks, attending sex parties or social gatherings and using recreational drugs during sex [9,10,21]. Indeed, men who have sex with men (MSM) have been disproportionately affected, with locally acquired community transmission predominant in all the affected countries. Moreover, concomitant sexually transmitted infections have been reported in the published international cohorts, with *N. gonorrhoeae*, *C. Trachomatis* and syphilis being the most common [1,2,3,18,19,20,21,22].

Based on these findings, the WHO still recommends avoidance of any sexual contact among mpox cases during the 21-day monitoring period, regardless of symptoms [9].

Mpxv can also be spread through respiratory secretions, although prolonged face-to-face contact may be required for transmission via this route [9,23]. Mpxv can also cross the placenta from the mother to fetus, leading to congenital mpox, although the rate of transmission or risk by trimester is unknown [24,25].

The incubation period for mpxv can range from 4 to 21 days. Invasive exposure through mucous involvement is linked to a shorter period of incubation than noninvasive transmission routes via droplets, characterized by 12–13 days [26].

The pathogenesis of mpox following skin inoculation seems to be similar to that of smallpox and other Orthopoxviruses, with evidence of viral replication in the skin and lymphatic system [27]. The formation of skin lesions starts with the vesicular stage, consisting of epidermal acanthosis and spongiosis, due to exocytosis of lymphocytes and neutrophils. The following apoptosis of keratinocytes and accumulation of intercellular fluid in the center of the lesion produce the vesicle. Then, the lesion starts to develop into a pustule containing a mixed inflammatory infiltrate of lymphocytes, eosinophils and neutrophils with apoptotic debris and few viable keratinocytes. Finally, the pustule becomes desiccated and forms a crust [28,29].

Mpxv infection stimulates the adaptative immune response, involving activated effector CD4+ and CD8+ T-cells, neutralizing antibodies (IgM, IgA and IgG), and the production of Th1-inflammatory cytokines (interferon gamma (IFN-γ), interleukin [IL]-1ra, IL-6, IL-8, and tumor necrosis factor (TNF)). This immune response contains viral replication and induces immunity in recovering patients [27,28,29].

The majority of mpox cases experience mild to moderate symptoms, followed by complete recovery with supportive care [30]. More than 90% of survivors have no complications and the most common sequelae are disfiguring scarring of the skin. Higher fatality and more severe clinical presentations have been reported in immunosuppressed patients, young adults and children, with mortality of 1–10%. Relevant changes regarding clinical features and complications have been followed by the emergence of vaccines [9,10,30].

Currently, in February 2023, the total number of confirmed cases of mpox in Italy is 957. During this outbreak, Lombardy was the region with the highest number of cases, followed by Lazio and Emilia-Romagna [31,32]. In this scenario, our open-access Sexual Health Clinic became an important reference center for mpox diagnosis, follow-up and treatment. The aim of this observational analysis is to describe demographical data, symptoms presentation and clinical course until outcome of individuals diagnosed with mpox, from May to October 2022, in IRCCS San Raffaele Hospital in Milan, Italy.

## 2. Materials and Methods

Our open-access Sexual Health Clinic is mainly dedicated to individuals who want to receive STI tests for periodical screening, or because of the presence of suspected symptoms, HIV pre-exposure prophylaxis users, and people who need to take HIV post-exposure prophylaxis.

At first access, we considered as suspected diagnosis of mpox every individual who presented symptoms that could be consistent with mpxv infection. The most suspicious symptom was the presence of cutaneous and/or mucosal lesions, deep-seated and well-circumscribed, often with central umbilication, characterized by progression through specific sequential stages from macules, papules, vesicles and pustules to scabs.

We also considered those who met the epidemiological criteria and had a high clinical suspicious for mpox. According to the Centers for Disease Control and Prevention (CDC)’s Case Reporting Recommendations for Health Departments, epidemiological criteria included a “history of close, intimate contact with people with a similar appearing rash or who received a diagnosis of confirmed or probable mpox or close or intimate in-person contact with individuals in a social experiencing mpox activity, including MSM or social event or traveled to a country endemic or with confirmed cases of mpox” [31].

In these cases, we performed physical examination and researched mpxv DNA via oropharyngeal, anal, genital and cutaneous swabs, plus blood, urine and seminal fluid. Approximately once per week, people repeated clinical evaluation and further research for mpxv on positive samples, based upon medical judgment.

In order to exclude concomitant sexually transmitted infections (STIs), we also completed a screening research for *Chlamydia trachomatis* and *Neisseria gonorrhoeae.* with real-time polymerases chain reaction (RT-PCR) on rectal and pharyngeal swabs and urines, plus serology tests for HIV, syphilis and hepatitis.

A total of 1836 samples were collected; in more detail, we acquired 590 swabs of cutaneous and mucosal lesions, 405 pharyngeal swabs, 326 samples of plasma, 229 of urine and 286 of seminal fluids. DNA was extracted using the QIA symphony DSP Virus/Pathogen Kit on QIA symphony SP instrument (QIAGEN—Milan, Italy). A RT-PCR assay, RealStar^®^ Orthopoxvirus PCR Kit 1.0 (altona DIAGNOSTICS—Milan, Italy), targeting variola virus and non-variola Orthopoxvirus species (*cowpox, monkeypox, raccoonpox, camelpox, vaccinia virus*) was used to detect the presence of non-variola DNA. Then, a specific RT-PCR targeting mpox virus DNA (Liferiver, SHANGHAI ZJ BIO-TECH CO.—Shanghai, China) was subsequently used for confirmation. Rectal, oropharyngeal and lesions swabs were collected by means of Universal Transport Medium, UTM-RT, swabs (COPAN Diagnostics—Brescia, Italy).

Urine and seminal fluid specimens were collected using sterile screw cap containers. Samples were considered positive with a Cycle Threshold (CT) value of RT-PCR ≤ 40. Laboratory analyses were performed at Laboratory of Clinical Microbiology, Virology and Bioemergencies of Luigi Sacco University Hospital, Milan, an Italian reference center for mpox diagnosis.

On their first visit, individuals filled out a questionnaire on their previous clinical history and high-risk sexual behaviors at time of diagnosis.

For people living with HIV (PLWH), we assessed the most recent CD4+ count cells/mm^3^, CD4+%, CD4+/CD8+ and HIV viral load, within the six months before mpox diagnosis.

We defined as complications the presence of tonsillar hyperemia with risk of airways obstruction, penile oedema causing phimosis/paraphimosis, severe proctitis and pain, with Numerical Rating Scales (NRS) score of at least 7–8.

Need for hospitalization was considered in case of worsening disease, not responsive to supportive therapy. For all hospitalized cases, we collected information on reason for admission, blood tests, use of antibiotic and/or antiviral therapies and hospitalization outcome.

This is an observational study. Frequencies and proportions were reported for categorial variables, as well as medians and interquartile ranges (IQR) for continuous variables. Data were analyzed with the use of Statistical Product and Service Solutions^®^ (SPSS^®^ Statistics, version 28.0 Mac Modified Release of November 2021, IBM^®^ Corp., Armonk, NY, USA).

## 3. Results

### 3.1. Main Baseline Characteristics

Overall, 140 individuals were diagnosed with mpox and included in this observational study. The baseline characteristics of all cases are summarized in Table 1.

Median age was 37 (IQR 33, 43) years old. Males were 137 (98%) and men who have sex with men (MSM) were 134 (96%). Ten people (7%) were Hispanic and the other 130 were Caucasian (93%).

As risk factors, we identified travels abroad in 49 (35%) individuals and previous close contact with known mpox cases in 49 (35%).

Smoking habit was reported in 52 (37%) cases and alcohol use in 55 (39%). Thirty-eight (27%) individuals referred to assumption of recreational substances, commonly referred to as “chemsex” drugs, such as methamphetamine and mephedrone, gamma-hydroxybutyrate (GHB) and gamma-butyrolactone (GBL).

Twenty (14%) individuals were previously vaccinated against smallpox during childhood, because smallpox vaccination was compulsory in Italy until 1977.

There were 66 (47%) people living with HIV (PLWH) and laboratory data are available in 55 (83%) of them, as illustrated in Table 2. Forty-three (31%) individuals assumed HIV pre-exposure prophylaxis (PrEP).

The types of sexual practices were classified as follows: vaginal-insertive sex mentioned in 1 (1%) case, vaginal-receptive sex in 3 (2%), anal-insertive sex in 70 (50%), anal-receptive sex in 75 (54%), oral-insertive sex in 79 (56%) and oral-receptive sex in 82 (59%).

### 3.2. Previous and Concomitant STIs

Previous and concomitant laboratory investigation confirmed that STIs were common (see Figure 1); 97 (69%) individuals had experienced at least one of *C. trachomatis, N. gonorrhoeae* and syphilis during their lifetime.

At the moment of mpox diagnosis, we observed *N. gonorrhoeae* in 18 (13%) cases, syphilis in 14 (10%) and *C. trachomatis* in 12 (9%). Two people (1%) were diagnosed with concomitant HIV infection.

### 3.3. Clinical Presentation

In our cohort, the most common symptoms related to mpox were fever (82; 59%), lymphadenopathy (80; 57%), mostly inguinal and cervical, proctitis (54; 39%), sore throat (31; 22%) and generalized rash (7; 5%), often associated with itchy skin. Figure 2 summarizes the clinical presentation of mpox in all 140 individuals.

The variety of lesions were represented especially by the cutaneous (108; 77%), but we also observed a particular predisposition of some anatomical sites such as genital (59; 42%), anal (47; 34%) and oral (36; 26%) areas. Among the genital lesions, a vulvar ulcer of was also included for one female of the cohort. The number of lesions varied from one to more than 50.

Acute proctitis (54; 39%) was manifested as rectal pain and tenesmus or purulent discharge. Other symptoms possibly related to proctitis were also painful defecation, fecal urgency, rectal bleeding and abdominal pain.

Tonsillitis showed such manifestations as sore throat (31; 22%) or trouble swallowing, with acute enlargement and reddening of the tonsils.

Moderate to severe penile oedema (5; 4%) was characterized by the swelling of the penile glans, often with retracted foreskin, who could not be returned to its anatomic position (e.g., paraphimosis).

Further minor clinical features included malaise, myalgia, diarrhea and urethritis. In particular, urethral involvement led to dysuria, difficulty urinating or hematuria.

One (1%) person manifested unilateral blepharoconjunctivitis and subconjunctival nodules as ocular involvement, without disturbance of vision, and followed by complete resolution.

Among hospitalized people and those presenting negative clinical conditions or complications (a total of 35, 25%), blood samples were collected to determine a complete blood count (CBC) and a biochemistry panel, including C-reactive protein (CRP), creatinine, electrolytes and liver function tests with alanine aminotransferase (ALT) and aspartate aminotransferase (AST). There were unspecified laboratory findings in 25 (71%) individuals, mainly characterized by increased levels of CRP with a median value of 30.1 mg/L (IQR 7.8–54.6) and abnormal transaminases with ALT > 60 U/L in 4 (11%) people.

### 3.4. Complication and Hospitalization

We observed 21 complications (15%) and hospitalization became necessary for nine individuals (6%) due to diseases unresponsive to supportive therapy, with a median hospital stay of 6 (IQR 3,7) days.

The spectrum of complications comprehended tonsillar hyperemia with risk of airways obstruction (3%), penile oedema causing paraphimosis (five cases, 4%), severe proctitis (11%) and negative clinical conditions (10%) mainly related to fever, malaise and severe pain, with Numerical Rating Scales (NRS) score of 7–8. All paraphimosis cases were resolved after manual reduction, with no surgical interventions required.

### 3.5. Treatment

Management of mpox involves supportive care and may also comprehend antibiotic and antiviral therapy for selected people.

Forty-five (32%) individuals needed anti-inflammatory drugs, such as non-steroidal anti-inflammatory drugs (NSAIDs), paracetamol or mesalamine in oral or rectal formulation. In some circumstances, prescription pain medications such as gabapentin have been used for short-term management of severe pain not controlled with other first-line treatments. Lidocaine ointment was provided for the perianal area for comfort and pain management with good effect. Furthermore, macrogol was used for pain support for stooling. Oral antihistamines afforded some relief for pruritis associated with mpox lesions and rash.

Thirty-six (26%) people were treated with antibiotic drugs, most often due to concomitant STIs, including azithromycin and ceftriaxone, penicillin and doxycycline. Amoxicillin/clavulanate was the therapy choice in the one case (1%) of bacterial superinfection of skin lesions.

In case of severity or worsening, eight (6%) individuals received treatment with antivirals. Among them, four were treated with intravenous injection of cidofovir 5 mg/kg plus oral administration of probenecid 2 g three hours before, then 1 g two and eight hours later. The other four took tecovirimat 600 mg orally twice a day for 14 days.

No individuals manifested side effects to antiviral treatment with cidofovir or tecovirimat. No significant changes in creatinine levels were detected and there was no onset of new symptoms after cidofovir administration.

### 3.6. Clinical Outcome and Weekly Trend of Mpox Diagnosis

One (1%) person was lost to follow-up. There were no deaths in our cohort and we declared full clinical recovery for the other 139 (99%) individuals, after a median duration of 18 (IQR 13–24) days of illness.

After the initially high incidence of mpox diagnosis with a peak in mid-July 2022, we moved to a sensible reduction from September 2022, as illustrated in Figure 3.

### 3.7. People Living with HIV (PLWH)

Laboratory data were available in 55 (83%) of 66 PLWH within the six months before mpox diagnosis, presenting a good immune-virological status with a median CD4+ cell count of 704 cells/mm^3^ (IQR 590, 953), as shown in Figure 4.

We observed HIV RNA < 200 copies/mL in 52 (95%) individuals and HIV RNA > 1500 copies/mL in the other three (5%), including the two with concomitant infection of HIV and mpxv.

At mpox diagnosis, 64 (97%) regularly received antiretroviral therapy, except for the two (3%) individuals who received concomitant diagnosis of HIV infection.

The median duration of illness was 17 (IQR 11-24) and the clinical manifestations were overlapping to those without HIV. Especially, as illustrated in Figure 2, they experienced lymphadenopathy (70%), fever (62%), proctitis (44%), sore throat (23%), generalized rash (8%) and others (58%), such as malaise, myalgia, diarrhea and urethritis. The lesions were cutaneous (76%), anal (48%), genital (45%) and oral (24%).

One (2%) person was treated with tecovirimat 600 mg orally twice a day for 14 days. Two (3%) people were treated with cidofovir 5 mg/kg plus oral administration of probenecid 2 g three hours before, then 1 g two and eight hours later.

Among them, we observed six (9%) cases of hospitalization due to bad clinical conditions, severe proctitis and tonsillar hyperemia with risk of airways obstruction, with median durations days of 5 days (IQR 3-8).

### 3.8. Vaccination against Smallpox or Mpox

Twenty (14%) individuals had been previously vaccinated against smallpox during childhood, because smallpox vaccination was compulsory in Italy until 1977. Also, other three people (2%) had recently been vaccinated with one dose of mpox vaccination and were scheduled to receive a second one after 28 days. Overall, 22 (96%) were MSM, with one (4%) transgender woman.

Among them, one (5%) person needed to be hospitalized because of a severe proctitis with bad clinical conditions and discharged after 7 days. During the hospitalization, he was treated with paracetamol and amoxicillin/clavulanate for 6 days with a moderate resolution of the symptoms.

One (5%) person experienced high fever and generalized and itchy rash, responsive to oral antihistamine drugs, associated with 32 lesions involving skin, genital and anal mucosal.

Six (27%) individuals presented concomitant STIs, in particular we observed two cases of *N. gonorrhea* infection, one case of syphilis, one coinfection of *C. trachomatis* plus syphilis and two coinfections of *N. gonorrhea* plus syphilis. They were all treated with antibiotics in order to resolve the concomitant STIs.

## 4. Discussion

This observational analysis shows that the main characteristics of all 140 individuals with mpox in our series reflect multiple facets of other international cohorts in this current outbreak [1,2,3,4,18,22].

Firstly, sexual transmission was most frequent, even though we cannot exclude the possibility of other ways of contracting the disease. With regards to this finding, many cases of mpox involved those who had joined in International Pride parades and large gatherings, reporting multiple sexual partners in the previous two weeks. Genital and anal lesions were common, at 42% and 34% respectively, and one (1%) female also presented a vulvar lesion. As further proof, concomitant STIs were also common (see Figure 1), in particular *C. trachomatis, N. gonorrhoeae,* and syphilis, underlining the importance of testing for STIs at mpox diagnosis [33,34,35,36].

Symptoms were heterogenous, not only associated with the presence of cutaneous and/or mucosal lesions, but also characterized by systemic involvement and possible evolution to severe disease. Fever and lymphadenopathy were the most frequent systemic manifestations since clinical presentation. Acute proctitis with rectal pain, tenesmus and purulent discharge played a role in the diagnostic process, due to its high frequency. It was also the symptom most needed to be treated with anti-inflammatory drugs and pain relievers. In addition, a case of unilateral blepharoconjunctivitis with subconjunctival nodules was observed, as a result of dissemination through direct contact with mpox lesions [37].

Complications were experienced by some individuals in our cohort, in particular tonsillar hyperemia with risk of airways obstruction, penile oedema causing phimosis/paraphimosis and significant bacterial superinfection. However, all symptoms were self-resolving and responsive to therapy, only few admissions to hospital were necessary, and there were no deaths in our series.

The typical clinical description of mpox starts from an initial prodromal phase with variable influenza-like symptoms, followed by the development of widespread, characteristic skin lesions on days 1–3 of illness. These can be pruritic, painful, indurated and umbilicated, generally with a diameter of 5 mm–1 cm. The macules uniformly progress over 1–2 days to papules, then to vesicles containing clear liquid and finally to pustules filled with yellow fluid, which then crust over and form scabs (days 7–14). Actually, in our experience, mpox can be a challenging diagnosis: the typical lesions, deep-seated and well-circumscribed, often with central umbilication, can be replaced by those with different appearance, emulating other diseases more commonly encountered in clinical practice (e.g., secondary syphilis, herpes and varicella zoster).

Regarding whether underlying immune deficiencies may lead to worse outcomes, in our cohort, among PLWH regularly receiving antiretroviral therapy and presenting a good immune-virological status, clinical presentation seemed to be similar to those without HIV in terms of symptoms’ characteristics, illness duration and clinical complications. Certainly, this finding must be corroborated by statistically significant data and further studies are needed to make a meaningful comparison between PLWH and without HIV in our cohort.

According to CDC current data, about 40% of those diagnosed with mpox in the United States also had HIV. Limited data suggest that PLWH, especially with low CD4 counts (<350 cells/mL) or detectable viral load, are at increased risk of severe mpox, hospitalization or even death if they are infected, compared to those without HIV [31]. However, it is still unknown if having HIV increases the likelihood of becoming sick with mpox if exposed to the virus. In addition, antiviral drugs used to treat mpox have minimal interactions with antiretroviral therapy and with common immunosuppressive medications [31,38].

In this study, even individuals who had previously received a smallpox vaccination during their youth received a diagnosis of mpox. The median age was over 50 years, so after this time neutralizing antibodies could progressively lose their ability in cross-protection. We should also consider that among the cohort more than three-quarters were living with HIV and others were taking immunosuppressive agents for co-morbidities. According to this hypothesis, immunosuppression and HIV-related immune-senescence may play a role in contracting mpox, despite vaccination [38,39].

The correct approach to the clinical management of mpox includes both general supportive care and use of antivirals with activity against mpox, in case of severe disease [38,40,41,42]. Four severe cases were successfully treated with cidofovir [43]. Cidofovir is a monophosphate nucleotide analogue, that undergoes cellular phosphorylation to its diphosphate form and competitively inhibits the incorporation of deoxycytidine triphosphate (dCTP) into viral DNA by viral DNA polymerase. Cidofovir demonstrates in vitro activity against some DNA viruses, including the herpesviruses, adenovirus, polyomavirus, papillomavirus and poxvirus. The most important toxicity of the drug is renal, but this can be reduced by concomitant administration of saline solution and probenecid [38,40].

Four severe cases were successfully treated with Tecovirimat. This is an inhibitor of an Orthopoxvirus protein required for the formation of a particle that is essential for dissemination within an infected host. Treatment with tecovirimat has been shown to reduce symptoms and duration of illness. In the United States, it was approved for the treatment of smallpox in July 2018 [36]. The recommended dose of tecovirimat depends upon the patient’s weight; as an example, for those ≥40 kg to <120 kg, the dose is 600 mg (three capsules) every 12 h. The duration of treatment is typically 14 days [38,41,42].

Based on CDC’s Case Reporting Recommendations for Health Departments, the use of tecovirimat should be considered first in immunocompromised people, who may also require a longer course of therapy. The subsequent addition of other medications (e.g., cidofovir, intravenous varicella immune globulin) should be contemplated based on the clinical scenario.

Despite some progress, there is a need for further research efforts to better understand the efficacy of these antivirals and establish their role as first line therapies for severe mpox cases.

At the beginning of this mpox outbreak, physical isolation was recommended to all patients until symptoms resolved. Resolution of lesions was defined as the crusting and subsequent formation of healthy new skin. Sexual intercourse has been demonstrated to be a well-established means of transmission, so the WHO recommend the avoidance of sexual activity during the 21-day monitoring period and suggest the use of condoms for 12 weeks after recovery [9]. More needs to be known about levels of the virus in seminal fluid and potential infectivity during the period after the recovery.

Currently, the most recently updated guidelines from the WHO of 22 December 2022 provide different recommendations [44]. Quarantine or exclusion from work are not necessary during the contact monitoring period as long as no symptoms develop. During the 21 days of monitoring, asymptomatic mpox cases are encouraged to rigorously practice hand hygiene and respiratory etiquette and to prevent contact with immunocompromised people, children [45,46] or pregnant women. While the WHO continues to review evidence on the possible transmission prior to onset of symptoms, the exhortation for mpox cases to avoid any sexual contact with others during the 21-day monitoring period, regardless of their symptoms, is still effective. However, it is still unknown whether seminal fluids might represent a viral reservoir.

In our cohort, similar to the trend in other international cohorts [31,32,44], we assisted in a huge number of mpox diagnoses between June and August 2022, with a peak of incidence in mid-July. We then observed a sensible reduction from September 2022, leading to the absence of new cases of mpox infection from mid-October, as illustrated in Figure 3. This trend can be connected to many factors. Firstly, a worldwide vaccination campaign started on August 2022 with two vaccines (JYNNEOS and ACAM2000), in order to prevent the spread of mpxv. Perhaps we can also consider the reduced amount of international travel and sexual interaction associated with large gatherings, such as Pride events, once summer had ended.

ACAM2000 is a replication-competent smallpox vaccine that can only be used in select patients and is associated with more adverse events. In the United States, it is approved for the prevention of smallpox [47]. According to the ACAM2000 clinical trial experience, CDC does not recommend this vaccine in case of immunodepression, due to the increased risk of severe side effects, such as myocarditis and pericarditis [31].

The JYNNEOS vaccine is made from a highly attenuated, nonreplicating vaccinia virus and has an excellent safety profile, even in immunocompromised people and those with skin disorders. In Italy, JYNNEOS is approved in individuals age ≥18 years at high risk for infection and is made of two doses administered with intradermal injection of 0.1 mL (28–35 days apart) [32].

Primary preventive (pre-exposure) vaccination (PPV) refers to the vaccination of individuals at risk of exposure to mpxv infection. PPV is recommended for groups at high risk for exposure to mpox in the current multi-country outbreak, for example gay, bisexual or other MSM groups. Further categories considered at possible risk may include individuals with multiple sexual partners, sex workers, health workers at risk of repeated exposure, laboratory personnel working with Orthopoxviruses, or clinical laboratory and health care personnel performing diagnostic testing for mpox. Mass vaccination is not recommended against outbreak of mpox at this time and vaccination is not recommended for the general public.

Post-exposure preventive vaccination (PEPV) indicates the immunization against mpox of close contacts of cases to prevent the onset of disease or moderate disease severity. PEPV is recommended for contacts of cases, ideally within four days of first exposure (and up to 14 days in the absence of symptoms). Pregnant women and immunocompromised people may be at risk of developing more severe disease when infected with mpxv, so these populations should be offered vaccination as a priority, if exposed. In infants, children and adolescents affected by eczema and other skin conditions or immunocompromised, mpox can spread through close, personal, often skin-to-skin contact, more likely with household members. Comparing children in West and central Africa, a preponderance of evidence also suggests that chickenpox is a risk factor for contracting mpox [45,46]

Current available treatment and vaccines effective towards mpox might play a significant role in treatment and prevention, with significant benefits regarding current disease outbreak.

The limitations of this study are correlated to its observational nature; because it has a lower standard of evidence than experimental studies, it is more open to confounding bias and is not able to demonstrate causalities. In addition, the small size sample does not permit overly strong conclusions. There is also a paucity of data on PLWH. No standardized follow-up was applied in all individuals and sampling was repeated for previous positive biological material upon clinical condition and medical judgment. No standardized treatment regimens were applied in all individuals and the decision whether to administer therapies was at the discretion of the treating physician.

## 5. Conclusions

Similarly to other international cohorts, sexual transmission was the most frequent mode, and concomitant STIs were common. Symptoms were heterogenous, self-resolving and responsive to therapy. Hospitalization was necessary in few patients.

There is uncertainty about the future development of mpox. The correct strategies to control a mpox outbreak must take into account important features such as the source of infection, all possible transmission routes and, last but not least, accurate communication in order to reach the public and affected communities and to develop an understanding of mpox. This would undoubtably help individuals at risk of infection, protecting their health and the health of their communities. Unfortunately, developing communication policies aimed at increasing awareness regarding mpox may represent an unmet need in many countries. In the United States of America, the CDC is constantly supporting healthcare providers and public health partners with information regarding the status of mpox outbreak, such as who is being diagnosed, what mpox lesions looks like and how to prevent or manage the illness.

It is also required to consider that specific behaviors predispose individuals to contract mpox. Specifically, gay, bisexual, and other MSM groups represent the largest proportion of cases described for the current outbreak, leading to a possible stigma associated with mpox, that it is mandatory to fight [48]. Therefore, the CDC also aimed to amplify its messaging on how mpox can be transmitted in multiple ways, recognizing the significance of sexual activity as a way of transmission, in order to educate sexually active populations on how best to keep healthy during the current outbreak.

It is fundamental to reach any disproportionately affected community without alarmist messages, giving fact-based messaging about mpox that provides people with appropriate tools to protect themselves and others. Messages are well accepted when delivered by partners or other trusted messengers, and they should be clear and non- judgmental. Any chance of stigma focused on sexual practice or community must be avoided, ensuring that the messages’ content is not homo-/bi-/transphobic or heterosexist.

Further studies addressing many different aspects of mpox, in particular regarding disease epidemiology, potential reservoirs, other possible means of transmission and predictors of severe disease are still needed.

## Figures and Tables

**Figure 1 viruses-15-00667-f001:**
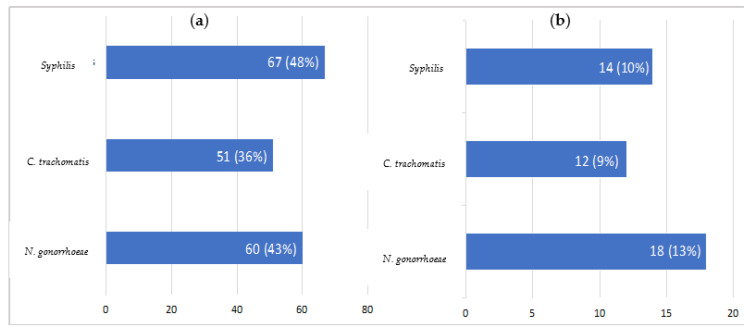
Previous (**a**) and concomitant (**b**) STIs.

**Figure 2 viruses-15-00667-f002:**
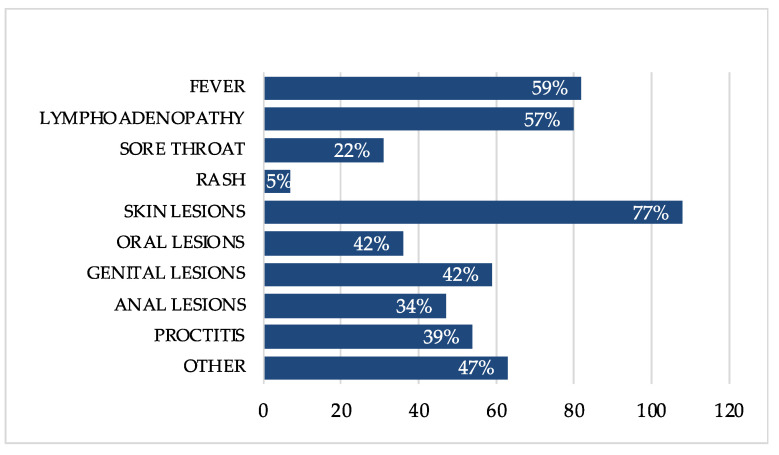
Clinical presentation of mpox in all 140 individuals.

**Figure 3 viruses-15-00667-f003:**
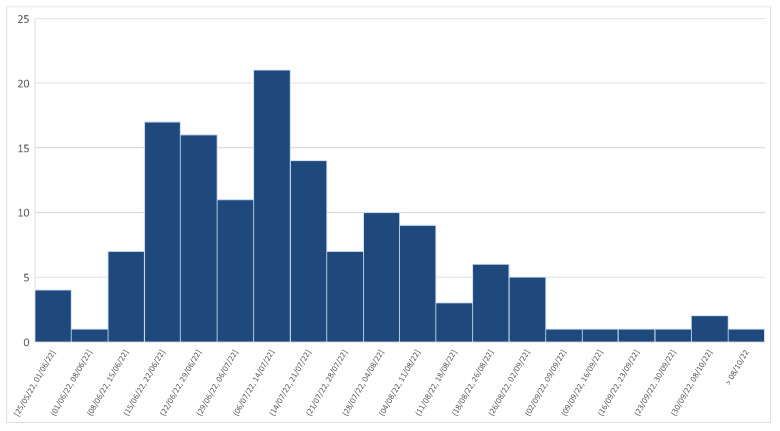
Weekly trend of mpox diagnoses from May to October 2022.

**Figure 4 viruses-15-00667-f004:**
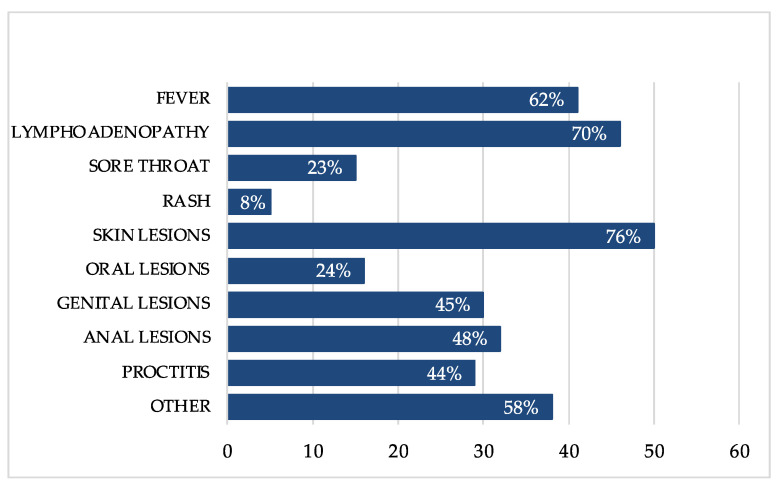
Symptoms of mpox in people living with HIV (PLWH).

**Table 1 viruses-15-00667-t001:** Patients’ descriptive characteristic.

Variables	N = 140
Age (median, IQR)	37 (33, 43)
Male (n, %)	137 (98%)
MSM (n, %)	134 (96%)
Travel abroad (n, %)	35 (25%)
Close contact with mpox cases (n, %)	49 (35%)
Previous smallpox vaccination (n, %)	20 (14%)
HIV (n, %)	66 (47%)
HCV (n, %)	10 (7%)
HBV (n, %)	1 (1%)
Immunodepression other than HIV (n, %)	3 (2%)
PrEP (n, %)	43 (31%)

**Table 2 viruses-15-00667-t002:** Laboratory values in PLWH diagnosed with mpox.

Variables	N = 55 *
HIV RNA < 50 copies/mL (n, %)	52 (95%)
CD4+ cell/mm^3^ (median, IQR)	704 (590, 953)
CD4+ % (median, IQR)	35 (30, 40)
CD4+/CD8+ (median, IQR)	0.98 (0.64, 1.23)
NADIR CD4+ cell/mm^3^ (median, IQR)	441 (249, 576)

* Laboratory data are available in 55 (83%) of 66 patients with HIV infection within six months before mpox diagnosis.

## Data Availability

Recorded data were anonymized and managed according to the Good Clinical Practice. Informed consent was obtained from all individual participants included in the study. Publication consent was obtained from all individual participants included in the study.

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
