# Peer review of "Human Monkeypox Experience in a Tertiary Level Hospital in Milan, Italy, between May and October 2022: Epidemiological Features and Clinical Characteristics"

_viruses, 2023, doi:10.3390/v15030667_

Round 1
Reviewer 1 Report
Candela et al. present clinically relevant reporting on human monkeypox incidence in an Italian tertiary hospital between May and October of 2022 during the peak spread of MPX in Europe. Whilst the paper is largely descriptive/reportative, it offers a significant hallmark of MPX infection and would be of significant use for future analysis.
Minor suggestions:
1. There are typographical and grammatical errors throughout the document.
Line
13: outbreaks
14: declare to declared
15: (The) Aaim of this....
15: Demographics data to demographical data
42: "with amount of 80 deaths, across 110 countries..."
43: "compared to ^(the) week from..."
54: "IIb, it is less virulent..."
55: "The current mpox epidemic seems (global notifications suggest?) to have originated in (from) the West Africa clade"
81: gonorrhea, chlamydia and syphilis should be captalised and include spp. after genus
97-101: Very limited detail on the immunology of MPX infection. This paragraph could be expanded.
98: "IgM (, IgA) and IgG)"
99: gamma interferon changed to interferon gamma
99" Define IL as interleukin in the first case
99: Define tumour necrosis factor (TNF)
103: "Ballooning" is not scientific
112: "Most reported deaths (have) occurred"
117: (The) Aaim of this observational...."
118: "course (un)till outcome"
145: formatting check
147: mm3
155: (The) Mmedian age...
160: "." at start of paragraph should be removed
204: CD4+
233: "three hour(s) before"
235: "analogue"
238: in vitro should be in vitro
Table 2: superscript mm3 to mm3 and add + to CD4 and CD8
287: Syntax of the sentence is confusing
290: CD4+
Author Response
Response 1. Thank you for these comments. The typographical and grammatical errors that you have mentioned have been revised. The corrections in the new manuscript are marked up using the “Track Changes” function.
- 97-101, now lines 514-518: we acknowledge that the immunological features represent a very important aspect of this specific disease. However, it is beyond the scope of the following study. We revised the manuscript according to the comments suggested by the reviewer 3 and 4 and we reduced the introduction. We believe that this modification might lead to a manuscript easier to follow for the readership.
- 103, now line 508: The term “ballooning degeneration” has been modified with “apoptosis”.
- 287, now lines 1234-1246: The sentence has been reformed as following: “Whether if underlying immune deficiencies may lead to worse outcomes, in our cohort among PLWH regularly receiving antiretroviral therapy and presenting a good immune-virological status, the clinical presentation seemed to be similar to those without HIV in terms of symptoms’ characteristics, illness duration and clinical complications. For sure, this finding must be corroborated by statistically significant data and further studies are needed to make a right comparison between PLWH and without HIV in our cohort.
According to CDC current data, about 40% of people diagnosed with mpox in the United States also had HIV. Limited data suggest that PLWH, especially with low CD4 counts (<350 cells/ml) or detectable viral load, are at increased risk of severe mpox, hospitalization or even death if they get the infection than people without HIV. However, it is still unknown if having HIV increases the likelihood of getting sick with mpox if exposed to the virus. Also, antiviral drugs used to treat mpox have minimal interactions with antiretroviral therapy and with common immunosuppressive medications [38]”.
Reviewer 2 Report
Journal Viruses (ISSN 1999-4915)
Type Article
Manuscript ID: viruses-2159123-
General Comments:
The article ' Human monkeypox experience in a tertiary level hospital in Milan, Italy, between May and October 2022: epidemiological features and clinical characteristics ' has been reviewed. The authors conducted an observational study to discuss the symptoms and clinical course of the Monkeypox (Mpox) virus in people living with HIV (PLWH) and without HIV in Milan, Italy. Overall, this does not provide particular information in this field.
Major Comments:
I have the following concerns.
Current data suggest that people diagnosed with Mpox around the world also have HIV and the Mpox outbreak has spread to locations that normally do not experience Mpox disease. The epidemiological features, clinical characteristics, clinical course, and outcome were well known. However, are there some knowledge gaps that need to be discussed? e.g., Are people with HIV more likely to get Mpox? Or Are People with Serious Immunocompromise (like advanced or untreated HIV) at increased risk of severe Mpox, or even death? Is the Mpox vaccine recommended and safe for PLWH? Did PLWH in antiretroviral drugs (ARV) support the benefits of protecting themselves from Mpox? Are people with Mpox infection treatment safe for PLWH? Given that the study lacks the main conclusion (similar to other international cohorts), it was suggested to compare the difference between people living with HIV (PLWH) and without HIV.
Minor Comments:
1. Please clearly state the primary outcome among people with Mpox infection.
2. Review the phrase 'Mpox virus infection', 'Mpxv', 'Mpox virus', and 'mpox infection' throughout the manuscript.
3. In Materials and Methods, there is a lack of a statistical method.
4. In the results, please, the data presentation focuses on the statistically significant results.
5. There are a lot of blurred paragraphs in the discussion that needs to be substantially revised given the focus on the most important finding in the tables to make it readable. I suggest rephrasing this a bit to make it clearer and to add some references. (Pages 7~9, lines 287 ~357)
6. Explain the limitations of this study. (Page 9, lines 375 ~378)
7. In the discussion, there is a lack of concussions. (Page 9, line 378)
8. The title should clearly state the objective of the study and focus on the Monkeypox (Mpox) virus in PLWH. such as 'The clinical course of the monkeypox virus among people living with HIV in Milan, Italy: A retrospective study '
9. The conclusion of this study should be consistent in the section Abstract and Conclusion.

Author Response
Response2.
Major comments: We thank the reviewer for these thoughtful comments. In the reviewed manuscript we tried to respond to these questions, comparing PLWH with those without HIV, in order to underly any differences or similarities.
In results, at lines 1050-1068, we added a paragraph regarding PLWH, as following:
“3.7. People living with HIV (PLWH)
Laboratory data were available in 55 (83%) of 66 PLWH within six months before mpox diagnosis, presenting a good immune-virological status with a median CD4+ cell count of 704 cells/mm3 (IQR 590, 953), as shown in figure 3. We observed HIV RNA <200 copies/mL in 52 (95%) individuals and HIV RNA >1500 copies/mL in the other 3 (5%), including those 2 with the concomitant infection of HIV and mpxv.
At mpox diagnosis 64 (97%) of them regularly got antiretroviral therapy, except for the 2 (3%) individuals who received concomitant diagnosis of HIV infection.
The median duration of illness was 17 (IQR 11-24) and the clinical manifestations were similar to those without HIV. Especially, as illustrated in Figure 2, they experienced lymphadenopathy (70%), fever (62%), proctitis (44%), sore throat (23%), generalized rash (8%) and others (58%), such as malaise, myalgia, diarrhea, urethritis. The lesions were cutaneous (76%), anal (48%), genital (45%) and oral (24%).
One (2%) person was treated with tecovirimat 600 mg orally twice a day for 14 days. Two (3%) people were treated with cidofovir 5 mg/kg plus oral administration of pro-benecid 2 g three hours before, then 1g two and eight hours later.
Among them, we observed 6 (9%) cases of hospitalization due to bad clinical con-ditions, severe proctitis and tonsillar hyperemia with risk of airways obstruction with median durations days of 5 days (IQR 3-8).”
Also, in discussion, the reviewed manuscript now reads as follow from line 1234 to 1246:
“Whether if underlying immune deficiencies may lead to worse outcomes, in our cohort among PLWH regularly receiving antiretroviral therapy and presenting a good immune-virological status, the clinical presentation seemed to be similar to those without HIV in terms of symptoms’ characteristics, illness duration and clinical complications. For sure, this finding must be corroborated by statistically significant data and further studies are needed to make a right comparison between PLWH and without HIV in our cohort.
According to CDC current data, about 40% of people diagnosed with mpox in the United States also had HIV. Limited data suggest that PLWH, especially with low CD4 counts (<350 cells/ml) or detectable viral load, are at increased risk of severe mpox, hospitalization or even death if they get the infection than people without HIV. However, it is still unknown if having HIV increases the likelihood of getting sick with mpox if exposed to the virus. Also, antiviral drugs used to treat mpox have minimal interactions with antiretroviral therapy and with common immunosuppressive medications [38].”
Then, at lines 1612-1615: “JYNNEOS vaccine is considered safe for PLWH. On the contrary, according to the ACAM2000 clinical trial experience, CDC does not recommend the ACAM2000 vaccine in case of immunodepression, due to the increased risk of severe side effects, such as myocarditis and pericarditis [Nalca A, Zumbrun EE. ACAM2000: the new smallpox vaccine for United States Strategic National Stockpile. Drug Des Devel Ther. 2010 May 25;4:71-9. doi: 10.2147/dddt.s3687. PMID: 20531961; PMCID: PMC2880337].”
Minor comments.
1, 3 and 4. We thank the reviewer for the thoughtful comments. However, it should be emphasized that the scope of the following study is to report demographical data, symptoms presentation and clinical course until outcome of patients diagnosed with mpox in our Center. Therefore, no group comparison or any kind of statistical analyses for event probability were performed. Indeed, it is beyond the scope of the following manuscript to report and eventually discuss statistically significant results. In consequence, the materials and methods section lacks statistical methodology description to report. In the reviewed manuscript, at lines 789-791: “This is an observational study. Frequencies and proportions were reported for categorial variables, as well as medians and interquartile ranges (IQR) for continuous variables. Data were analyzed with the use of SPSS software, version 28 (IBM).”
- All the typographical and grammatical errors that you have mentioned have been revised. The corrections in the new manuscript are marked up using the “Track Changes” function.
- All the discussion has been reviewed, according to the comments suggested by the reviewer. We believe that this modification might lead to a manuscript easier to follow for the readership. The Figure 2, illustrating the symptoms of mpox in people living with HIV (PLWH), has also been added.
- In discussion, the reviewed manuscript now reads as follow from line 1601 to 1609: “The limitations of this study are correlated to its observational nature, because it has a lower standard of evidence than experimental ones, it is more open to confounding bias and it isn’t able to demonstrated causalities. In addition, the small size sample doesn’t permit to make overly strong conclusions. There is a paucity of data on PLWH too. No standardized follow-up was applied in all individuals and sampling was repeated for the previous positive biological material upon clinical condition and medical judgment. No standardized treatment regimens were applied in all individuals and the decision whether to administer therapies was discretion of the treating physician”.
7 and 9. The Conclusions part has been added in the main text (from line 1610 to 1613) and in the abstract (at lines 33-37), as following: “CONCLUSIONS: Similarly to other international cohorts, sexual transmission was the most frequent and concomitant STIs were common. Symptoms were heterogenous, self-resolving and responsive to therapy. Hospitalization was necessary in few patients. There is uncertainty about the future development of mpox and further studies (e.g. potential disease reservoirs, other possible ways of transmission, predictors of severe disease) are still needed.”
- We thank the reviewer for this comment. However, even if PLWH represent the 47% of our cohort, we believe that this modification is beyond the aim of the study to describe demographical data, symptoms presentation and clinical course till outcome of individuals diagnosed with mpox in our open-access Sexual Health Clinic, which is currently the largest cohort in Italy.
Reviewer 3 Report
This work presents a study about 140 individuals with a diagnosis of monkeypox infection in a hospital in Milan, Italy. This theme is of high interest for physicians, but there are important issues that limit the publication of this manuscript.
Thus, the authors should consider:
Abstract:
- the authors establish that “Initially, we performed physical examination and screening for sexual transmitted infections (STIs)”. Were the patients submitted to the Unit or they were diagnosed during routine evaluation?. This sentence led to the interpretation of active search of cases of MP. Please, clarify.
- Seminal fluid of patients? Why? Was this a prospective study about the finding of poxvirus in seminal fluid? (see ethical consideration)
- Why the authors consider that “travel abroad” is a risk factor for MP infection? Explain it.
- Conclusions: The authors stated that “concomitant STIs were common”, but no data are included about this fact in the results.
Introduction.
- Too long. I don’t believe that introduction should include a revision of MP infections (epidemiology, pathogenesis, immunology), since these data are not related with the aim of your work (and neither with your results).
Methods
- Again, the authors established “at first access to sexual clinic (line 122)”. Were the patients submitted to the clinic or they were searching for medical care because of symptomatology? As written, the finding of MP seems to be merely casual (“in case of symptoms related, we … (line 127). Better explain the reason to be attended and, in case of casual, how many patients had MP at the time of evaluation.
- Again, 286 seminal fluids were collected. I can’t imagine the reason for extracting seminal fluid in patients with cutaneous or rectal lesions.
- Line 145. A lost line? What questionnaire was performed?
Results
- Again, travel abroad was a risk factor for MP infection? Where and why?
- Why the authors included previous STIs? (line 176). At least that the authors considered the risk of previous STIs for having more concomitant STIs
- The presentation of symptomatology is not clear. Use Tables to know how many had proctitis,diarrhea, urethritis,…
- Line 204. The authors stated similar CD4 count in HIV positive and negative (better only mention that CD4 count were high in those with HIV infection)
- If including complications, a brief sentence about how was defined complications in the Methods section. Same for deciding hospitalization. Use Tables
- In the Treatment section, the first paragraph (line 218) is a superficial revision of the different possibilities of treatment. Better delete it. Same for including a general paragraph about cidofovir and tecovirimat, not related with your study.
Table 1. If included the rate of previous smallpox vaccination, the authors should clarify the meaning of this sentence (had history of being vaccinated in the 70’s or 80’s, or currently receiving smallpox vaccination in the months previously to MP infection?)
Figure 2. To reflect the weekly trend because of temporal changes of MP incidence? In that case, the authors should explain why they expected high or low incidences.
Discussion
- Except for some isolated sentences (concomitant STIs, 4 cases receiving cidofovir), the discussion is not addressed to the results of the work, neither to compare the findings with previous studies, or highlight special features. Thus, the authors included general information about MP infection.
Ethical
- Approval data of the Ethic committee December 2017 ?
Author Response
Response 3. Abstract:
- We thank the reviewer for the thoughtful comments. The suggested corrections are marked up in the new manuscript using the “Track Changes” function.
Our open-access Sexual Health Clinic is mainly dedicated to individuals who want to receive STIs tests for periodical screening or because of the presence of suspected symptoms, HIV pre-exposure prophylaxis users and people who need to take HIV post-exposure prophylaxis.
Among all of them, between May and October 2022, 140 people were diagnosed with mpox and included in this observational study.
The reviewed manuscript now reads as follow from line 18 to 21 “Among people who accessed to our Sexual Health Clinic, we considered as suspected diagnosis of mpox individuals with consistent symptoms and epidemiological criteria. Following the physical examination, oropharyngeal, anal, genital and cutaneous swabs, plus plasma, urine and seminal fluid were collected as biological materials to detect mpxv DNA. We also performed a screening for sexually transmitted infections (STIs).”
- We thank the reviewer for this thoughtful comment. As already mentioned, during this mpox outbreak, sexual transmission has been well-established to be most frequent way of infection, even though not the only one possible. At the beginning, WHO suggested to elude sexual activity during the 21-day monitoring period and suggested to use condoms for 12 weeks after they recovered. Currently, it’s still effective the exhortation to mpox cases to avoid any sexual contacts with others during the 21-day monitoring period, regardless of their symptoms. However, it is still unknown whether seminal fluids might represent a viral reservoir. In some cases, at discretion of the treating physician and obviously in concordance with patients, seminal fluid analysis were performed. We believe that the inclusion of such data might be of interest for the readership.
We based these considerations on the following manuscripts:
- Thornhill JP, Antinori A, Orkin CM. Monkeypox Virus Infection across 16 Countries - April-June 2022. Reply. N Engl J Med. 2022;387(25):e69. doi:10.1056/NEJMc2213969
- Peiró-Mestres A, Fuertes I, Camprubí-Ferrer D, et al. Frequent detection of monkeypox virus DNA in saliva, semen, and other clinical samples from 12 patients, Barcelona, Spain, May to June 2022. Euro Surveill. 2022;27(28):2200503. doi:10.2807/1560-7917.ES.2022.27.28.2200503
- Antinori A, Mazzotta V, Vita S, et al. Epidemiological, clinical and virological characteristics of four cases of monkeypox support transmission through sexual contact, Italy, May 2022. Euro Surveill. 2022;27(22):2200421. doi:10.2807/1560-7917.ES.2022.27.22.2200421
- Lapa D, Carletti F, Mazzotta V, et al. Monkeypox virus isolation from a semen sample collected in the early phase of infection in a patient with prolonged seminal viral shedding. Lancet Infect Dis. 2022;22(9):1267-1269. doi:10.1016/S1473-3099(22)00513-8
- Raccagni AR, Candela C, Mileto D, et al. Monkeypox infection among men who have sex with men: PCR testing on seminal fluids. J Infect. 2022;85(5):573-607. doi:10.1016/j.jinf.2022.07.022
- Rizzo A, Mileto D, Moschese D, et al. Role of multi-site sampling in the diagnosis of human Monkeypox. J Infect. 2023;86(2):154-225. doi:10.1016/j.jinf.2022.12.010
- We thank the reviewer for this thoughtful comment. As mentioned at lines 175-183: “the first outbreak of mpox in the United States in 2003 and the other further sporadic cases in western countries were mostly related to travel in endemic areas. On the other hand, the first case of mpox of this current global outbreak, reported in the United Kingdom in mid-May 2022, and the succeeding other cases worldwide were not associated with recent travel to endemic areas or close contact with a person known to have mpox, providing the evidence of a community spread”.
Based on this consideration, the CDC’s Case Reporting Recommendations for Health Departments suggest to define as suspected the cases that meet the following epidemiological criteria: “history of close, intimate contact with people with a similar appearing rash or who received a diagnosis of confirmed or probable mpox or close or intimate in-person contact with individuals in a social experiencing mpox activity, including MSM or social event or travelled to a country endemic or with confirmed cases of mpox” (https://www.cdc.gov/poxvirus/monkeypox/clinicians/case-definition.html).
- According to the thoughtful suggestion of the reviewer, in results, at lines 28-30, we added the following sentence, in order to provide the evidence of this conclusion: “At mpox diagnosis, we also observed N. gonorrhoeae in 18 (13%) cases, syphilis in 14 (10%) and C. trachomatis in 12 (9%). Two (1%) people received a concomitant diagnosis of HIV infection.”
Introduction:
- According to the reviewer suggestion, in harmony also with comments of other reviewers that manifested interest for this section of the manuscript, the introduction has been reviewed and reduced. In the revision processes, we tried to build a stronger connection between this new introduction and the discussion of our observational study.
Methods:
- See reply 1 of the section “Abstract”. The reviewed manuscript now reads as follow from line 610 to 630: “At first access, we considered as suspected diagnosis of mpox every individual who presented symptoms that could be consistent with mpxv infection. The more suspicious symptom was the presence of cutaneous and/or mucosal lesions, deep-seated and well-circumscribed, often with central umbilication, characterized by progression through specific sequential stages from macules, papules, vesicles, pustules to scabs. We also considered those who met the epidemiological criteria and had a high clinical suspicious for mpox. According to Centers for Disease Control and Prevention (CDC)’s Case Reporting Recommendations for Health Departments, in epidemiological criteria were included a “history of close, intimate contact with people with a similar appearing rash or who received a diagnosis of confirmed or probable mpox or close or intimate in-person contact with individuals in a social experiencing mpox activity, in-cluding MSM or social event or traveled to a country endemic or with confirmed cases of mpox” [38]. In these cases, we performed physical examination and researched mpxv DNA on oropharyngeal, anal, genital and cutaneous swabs, plus blood, urine and seminal fluid. Approximately every week, people repeated clinical evaluation and further research for mpxv on positive samples upon medical judgment. In order to exclude concomitant sexually transmitted infections (STIs), we also completed a screening research for Chlamydia trachomatis, Neisseria gonorrhoeae with real-time polymerases chain reaction (RT-PCR) on rectal and pharyngeal swabs and urines plus serology tests for HIV, syphilis and hepatitis.”
We also underlined the number of patients diagnosed with mpox at the time of evaluation, at line 800, as following: “Overall, 140 individuals were diagnosed with mpox and included in this observational study.”
- See comment #2 of Abstract session.
- All the typographical and grammatical errors that you have mentioned have been revised. The reviewed manuscript now reads as follow from line 720 to 721: “On their first visit, individuals filled out a questionnaire on their previous clinical history and high-risk sexual behaviors at time of diagnosis”.
Results:
- See comment #3 of Abstract session.
- We thank the reviewer for the thoughtful comments, giving us the opportunity to clarify information about this aspect of the study. The majority of the 140 patients diagnosed with mpox regularly attend to our open-access Sexual Health Clinic and were treated for previous STIs. As the reviewer said, we likely consider the history of previous STIs as risk factors to have more concomitant STIs. We believed that this information could be interesting, providing the importance of sexual activity as main way of transmission. Although it is mandatory to fight a possible stigma associated with mpox, it has been well-established that specific behaviors predispose individuals to contract the infection, especially having sex with multiples sexual partners, as described for the current outbreak.
- According to the reviewer suggestion, the new manuscript now includes the Figures 2 and 3, with the presentation of clinical manifestations of mpox in all 140 individuals and Symptoms of mpox in people living with HIV (PLWH). We also modified the manuscript, at line 783, as following: “The figure 2 summarizes the clinical presentation of mpox in all 140 individuals. The variety of lesions was represented especially by the cutaneous ones (108; 77%), but we also observed a particular predisposition of some anatomical sites such as genital (59; 42%), anal (47; 34%) and oral (36; 26%) area. Among the genital lesions is also included the vulvar ulcer of one female of the cohort. The number of lesions varied from one to more than 50. The acute proctitis (54; 39%) was manifested as rectal pain and tenesmus or purulent discharge. Other symptoms possibly related to proctitis were also painful defecation, fecal urgency, rectal bleeding and abdominal pain. Tonsillitis showed more manifestations as sore throat (31; 22%) or trouble swallowing with acute enlargement and reddening of the tonsils. The moderate to severe penile oedema (5; 4%) was characterized by the swelling of the penile glans, often with retracted foreskin who cannot be returned to its anatomic position (e.g., paraphimosis). Further minor clinical features included malaise, myalgia, diarrhea, urethritis. In particular, urethral involvement led to dysuria, difficulty urinating or hematuria. One (1%) person manifested unilateral blepharoconjunctivitis and subconjunctival nodules as ocular involvement, without disturbance of vision and followed by complete resolution [32].”
- According to the reviewer, line 204 has been reviewed. At line 987, we added a new paragraph 3.7. People living with HIV (PLWH) to better explain what we had observed in this population. We believe that this modification might lead to a manuscript easier to follow for the readership.
- In methods, at lines 727-731, we defined complication and the needing of hospitalization as following: “We defined as complications the presence of tonsillar hyperemia with risk of airways obstruction, penile oedema causing phimosis/paraphimosis, severe proctitis and pain with Numerical Rating Scales (NRS) score of 7-8 at least. Needing of hospitalization was considered in case of worsening disease, not responsive to supportive therapy.”
- According to the reviewer suggestion, the treatment session has been reviewed, at lines 828-862.
Table 1. The new manuscript new includes the paragraph “3.8. People who were vaccinated against smallpox or mpox” at lines 988-1060.
Figure 2 (now Figure 4). In results, at lines 1339-1343, the manuscript has been modified adding a description of the weekly trend of mpox diagnosis, as following: ”In our cohort, as similar to the trend of other international cohorts, we assisted to a huge number of mpox diagnosis between June and August 2022, with a peak of incidence in mid-July. Then, we also observed a sensible reduction of them from September 2022, leading to the absence of new cases of mpxv infection from mid-October, as illustrated in Figure 3. This trend can be connected to many reasons. Firstly, a worldwide vaccination campaign started on August 2022 with two vaccines (JYNNEOS and ACAM2000) in order to prevent the spread of mpox. Perhaps, we could also consider the less amount of international travel and sexual interactions associated with large gatherings, such as pride events, once summer ended.”
Discussion:
According to the reviewer suggestion, in harmony also with comments of other reviewers, the discussion part has been modified (page 9-11). We believe that this modification might lead to a manuscript easier to follow for the readership.
Ethical:
We thank the reviewer for the opportunity to clarify information regarding data collection and ethical approval. All the subjects gave their informed consent to be part of the Centro San Luigi (CSL) Cohort which included people with HIV or at high risk of acquiring HIV infection. The CSL-HIV Cohort was approved by the Ethics Committee of the San Raffaele Hospital, Milan, Italy (date of approval December 4, 2017, protocol no. 34). The written consent also included a statement regarding the didactic and scientific use of their clinical data and iconographic materials for publications, conferences, retrospective analysis and protocols.
Reviewer 4 Report
It is important for each country to document their experience with diagnosis and management of monkeypox. The differences between countries have been more than expected. In this manuscript, a medical group from Milan, Italy, carefully describe their mpox outbreak in sexually active adults. A few comments for improvement are listed below.
1. Introduction. It is unusual to comment much about the Introduction. But there have been hundreds of reports published about monkeypox in the medical literature over the past 2 years. There is no need for another long Introduction. Suggest a 50% reduction in length of Introduction between lines 50-116. Go more quickly to the main message.
2.Line 117. Add a sentence about the total number of cases of monkeypox reported to WHO that have occurred in Italy.
3. Results, line 244. Add the total number of patients who were prescribed cidofovir
4.Results, line 251. Add the total number of patients who were prescribed Tecovirimat.
5.Results, line 256. Add a new section to Results called Patients who were vaccinated against smallpox. Describe the course of monkeypox disease in the 22 patients who were previously vaccinated against smallpox. Did they have milder disease? Were any of the 22 placed in a hospital? This would become section 3.6 and current section 3.6 would become 3.7.
6. Discussion. Monkeypox in children, lines 345-347. There are 2 newly published articles in excellent journals about monkeypox in children. Both were published online in late December 2022. Both provide good insight into the disease in children. Please cite these 2 papers in References and remind physicians that cases of monkeypox in children are not sexually transmitted. (a) A. Beeson et al, Mpox in children and adolescents. PEDIATRICS (2023) 151 (2):e2022060179; and (b) O. Khallafallah et al, Reassessment of evidence about coinfection of chickenpox and monkeypox in African children. VIRUSES (2022) Dec 15; 14 (12):2800.
Author Response
Response 4: We thank the reviewer for these thoughtful comments. The suggested corrections are marked up in the new manuscript using the “Track Changes” function.
- As you suggested, the introduction has been reviewed and reduced.
- The sentence about the number of cases in Italy has been added, at line 547, as following: “Currently, in February 2023, the total number of confirmed cases of mpox in Italy is 957. During this outbreak, Lombardy was the region with the highest number of cases, followed by Lathium and Emilia-Romagna [11]. In this scenario, our open-access Sexual Health Clinic became an important reference center for mpox diagnosis, follow-up and treatment.”
3 – 4. At lines 1013-1019 the prescription of antiviral therapy is explained as following: “In case of severity or worsening, 8 (6%) individuals received treatment with antivirals. Among them, 4 people were treated with intravenous injection of cidofovir 5 mg/kg plus oral assumption of probenecid 2 g three hours before, then 1g two and eight hours later. The other 4 people took tecovirimat 600 mg per os twice a day for 14 days.”
- Among results, we have added a section at 3.6 regarding patients who were vaccinated against smallpox/mpox, as following:
“3.7. People who were vaccinated against smallpox or mpox
Twenty (16%) individuals were previously vaccinated against smallpox during childhood, because smallpox vaccination was compulsory in Italy until 1977. Other three (were recently vaccinated with one dose of mpox vaccination and were scheduled to receive a second one after 28 days. Overall, 22 (96%) were MSM, with 1 (4%) transgender woman. Among them, 1 (5%) person needed to be hospitalized because of a severe proctitis with bad clinical conditions and discharged after 7 days. During the hospitalization, he was treated with paracetamol and amoxicillin/clavulanate for 6 days with a moderate resolution of the symptoms. One (5%) person experienced high fever and generalized and itchy rash, responsive to oral antihistamine drugs, associated with 32 lesions involving skin, genital and anal mucosal. Six (27%) individuals presented concomitant STIs, in particular 2 N. gonorrhea, 1 syphilis, 1 coinfection of C. trachomatis and syphilis and 2 coinfections of N. gonorrhea and syphilis. They all were treated with antibiotic therapies in order to resolve the concomitant STIs.”
- These two interesting papers have been added to references and mentioned in the discussion, underlining the different ways of transmission of mpox in children. In discussion, the reviewed manuscript now reads as follow from line 1969 to 1973: “Also in infants, children and adolescents affected by eczema and other skin conditions or immunocompromised mpox can spread through close, personal, often skin-to-skin con-tact, more likely for household members. Between children in west and central Africa, a preponderance of evidence suggests also that chickenpox is a risk factor for contracting mpox [46, 47]”.
Reviewer 5 Report
Candela et al. have reported on their experience with the management of mpox cases. The work has merits, however, I have a number of suggestions on how it can be further improved.
1.) Introduction, lines 94-96 and discussion, lines 277-286: Is it really the authors’ experience that the skin lesions resembled the classical ones described for smallpox in the literature? To our experience, there was a high variability of lesions, making differential diagnosis from other infectious skin diseases quite challenging. Also, there was a more pronounced local reaction at the inoculation site with seemingly less systemic spreading to our experience. The authors might want to comment whether they have seen such phenomena as well.
2.) Introduction, lines 112-113) The authors should comment on potential clade-associations of the described high mortality. We have seen a lot of mpox cases and not even a single one of them died. Obviously, the authors’ experience was quite similar.
3.) Materials & methods chapter, second paragraph: The authors did not mention assessments of vaginal swabs or fluids for their STI screenings. Was this type of sample material simply forgotten in the description or was it really not assessed?
4.) Results, lines 180-182: The high proportions of localized sexual-transmission-associated lesions deserve further discussion in the discussion chapter, as this is a clear difference compared to earlier mpox outbreaks as well as former smallpox outbreaks.
5.) Results, line 200-201: It would be interesting to learn more about the mentioned “non-specific laboratory findings”.
6.) Results, lines 206-207 and later in the discussion: The authors should at least explain what they mean than speaking of a “sensible reduction of mpox diagnosis”.
7.) Results, line 233: “Oral assumption” should mean “oral administration”, I suppose?
8.) Results, lines 235-251: These paragraphs are discussion rather than results.
9.) Discussion, lines 270-271: The authors describe “sexual transmission” to be “not the only way”. Do they have any hints how transmission occurred in the other, non-sex-associated cases? Was close personal contact involved? This question is of high relevance, considering partly very strict preventive measures including isolation schemes in line with WHO recommendations. To our experience, very close contacts are necessary for a transmission, making a general need for isolation difficult to justify. The authors might comment on this, also on the position of some other authors that mpox should be considered as an STI due to its primary transmission on the sexual route.
10.) The authors partly use the old nomenclature “monkeypox” and partly the new nomenclature “mpox”. They should standardize this.
Author Response
Response 5. We thank the reviewer for the comments. The corrections in the new manuscript are marked up using the “Track Changes” function.
1. According to the comments suggested by the reviewer 2, 3 and 4, we revised and reduced the introduction. We believe that this modification might lead to a manuscript easier to follow for the readership. So, we decided to describe this aspect in discussion, underling the possibility of a challenging diagnosis. The reviewed manuscript now reads as follow from line 1513 to 1522: “The typical clinical description of mpox starts from an initial prodromal phase with variable influenza-like symptoms, followed by the development of widespread, char-acteristic skin lesions on days 1–3 of illness. They can be pruritic, painful, indurated and umbilicated, generally with diameter of 5 mm-1 cm. The macules uniformly progress over 1–2 days to papules, then to vesicles containing clear liquid and finally to pustules filled with yellow fluid, which then crust over and form scabs (days 7–14). Actually, in our experience, mpox can be a challenging diagnosis: the typical lesions, deep-seated and well-circumscribed, often with central umbilication, can be replaced by ones with dif-ferent appearance, emulating other diseases more commonly encountered in clinical practice (e.g., secondary syphilis, herpes and varicella zoster).”
2. We thank the reviewer for the considerate comments. However, it is beyond the scope of the following manuscript to report and eventually discuss the potential clade-associations of high mortality. This is a interesting thought, but this study is not able to propose any hypothesis. Talking about mortality, the reviewed introduction of the manuscript now reads as follow from line 581 to 587: “The majority of mpox cases experience mild to moderate symptoms, followed by complete recovery with supportive care. More than 90% of survivors have no complications and the most common sequelae are disfiguring scarring of the skin. Higher fatality and more severe clinical presentations have been reported in immunosuppressed patients, young adults and children, with mortality of 1–10% [9, 10]. Relevant changes regarding clinical features and complications have been followed the emergence of vaccines. [9, 10].”
3. Vaginal swabs or fluids have not been accessed for all the female patients, excepting for the only one with a vulvar lesion (included in the category of genital lesions), as mentioned in results at line 839.
4 and 9. We thank the reviewer for this thoughtful comment. During the course of the manuscript we often underlined the important role of sexual contact as way of transmission. It’s required to consider that specific sexual behaviors predispose individuals to contract mpox, providing the evidence that gay, bisexual and other MSM represent the largest proportion of cases described for this current outbreak. We also reassumed at line 1173 to 1180: “ Firstly, sexual transmission was the most frequent [38, 39, 41], even though not we can’t exclude the possibility of other ways to contract the disease. On behalf of this finding, many cases of mpox interested people who had joined in international pride parades and large gatherings, reporting multiple sexual partners in the previous two weeks. Genital and anal lesions were common, 42% and 34% respectively, and one (1%) female also presented a vulvar lesion. As further proof, also concomitant STIs were common (see Figure 1), in particular C. trachomatis, N. gonorrhoeae, and syphilis, underlining the im-portance of testing STIs at mpox diagnosis [40-43].” However, it is mandatory to fight possible a stigma associated with mpox.
5. We thank the reviewer for the thoughtful comment. We added a more detailed explanation from line 855 to 861: “Among hospitalized people and those presenting bad clinical conditions or complications (a total of 35, 25%), blood samples were collected to determine a complete blood count (CBC) and a biochemistry panel, including C-reactive protein (CRP), creatinine, electrolytes and liver function tests with alanine aminotransferase (ALT), aspartate aminotransferase (AST). We assisted to unspecific laboratory findings in 25 (71%) individuals, mainly characterized by increased levels of CRP with a median value of 30.1 mg/L (IQR 7.8-54.6) and abnormal transaminases with ALT > 60 U/L in 4 (11%) people.”
6. In discussion, at line to 1388 to 1392, the manuscript has been reveiwedas following: “In our cohort, as similar to the trend of other international cohorts, we assisted to a huge number of mpox diagnosis between June and August 2022, with a peak of incidence in mid-July. Then, we also observed a sensible reduction of them from September 2022, leading to the absence of new cases of mpxv infection from mid-October, as illustrated in Figure 3.”
7 and 8. All these corrections have been applied, as suggested by the reviewer.
10. All the typographical and grammatical errors that you have mentioned have been revised. The new nomenclature with “mpox” and “mpxv” has been used in all the manuscript.
Reviewer 6 Report
The article is well written and provides a concise but pleasant review, particularly on the clinical and epidemiological aspects of cases of mpox in a tertiary-level hospital in MiIan. I was happy to read it; the text is written appropriately and contains basically all the information a clinician needs to know about the clinical aspects of the infection. The figures are informative and necessary for the development of the article.
I would suggest that the authors to check the sentence in lines 68-69 to avoid the repetition of "providing the evidence" .... lines 68-69: check: "providing the evidence 68 - of community spread providing the evidence of community spread..." 69
Author Response
Response: We thank the reviewer for the comment. All the typographical and grammatical errors that you have mentioned have been revised. The corrections in the new manuscript are marked up using the “Track Changes” function.
Round 2
Reviewer 2 Report
The revised version is improved and can be accepted.
All comments have been addressed.